# Minigo: A Case Study in Reproducing Reinforcement Learning Research

**Brian Lee, Andrew Jackson, Tom Madams, Seth Troisi, Derek Jones**
Google, Inc.
{brianklee, jacksona, tmadams, sethtroisi, dtj}@google.com

## Abstract

The reproducibility of reinforcement-learning research has been highlighted as a key challenge area in the field. In this paper, we present a case study in reproducing the results of one groundbreaking algorithm, AlphaZero, a reinforcement learning system that learns how to play Go at a superhuman level given only the rules of the game. We describe Minigo, a reproduction of the AlphaZero system using publicly available Google Cloud Platform infrastructure and Google Cloud TPUs. The Minigo system includes both the central reinforcement learning loop as well as auxiliary monitoring and evaluation infrastructure. With ten days of training from scratch on 800 Cloud TPUs, Minigo can play evenly against LeelaZero and ELF OpenGo, two of the strongest publicly available Go AIs. We discuss the difficulties of scaling a reinforcement learning system and the monitoring systems required to understand the complex interplay of hyperparameter configurations.

## 1   Introduction

In March 2016, Google DeepMind's AlphaGo [1] defeated world champion Lee Sedol by using two deep neural networks (a policy and a value network) and Monte Carlo Tree Search (MCTS) to synthesize the output of these two neural networks. The policy network was trained via supervised learning from human games, and the value network was trained from a much larger corpus of synthetic games generated by sampling game trajectories from the policy network. AlphaGo Zero[2], published in October 2017, described a continuous pipeline, which when initialized with random weights, could train itself to defeat the original AlphaGo system. The requirement for expert human data was replaced with a requirement for vast amounts of compute: approximately two thousand TPUs were used for 72 hours to train AlphaGo Zero to its full strength. AlphaZero[3] presents a refinement of the AlphaGoZero pipeline, notably removing the gating mechanism for publishing new models.

In many ways, AlphaGo Zero can be seen as the logical culmination of fully automating and streamlining the bootstrapping process: the original AlphaGo system was bootstrapped from expert human data and reached a final strength that was somewhat stronger than the best humans. Then, by generating new training data with the stronger AlphaGo system and repeating the bootstrap process, an even stronger system was created. By automating the bootstrapping process until it is continuous, a system is created that can train itself to surpass human levels of play, even when starting from random play.

In this paper, we discuss our experiences creating Minigo. About half of our effort went into rebuilding the infrastructure necessary to coordinate a thousand selfplay workers. The other half of the effort went into monitoring infrastructure to test and verify that what we had built was bug-free. Despite having at hand a paper describing the final architecture of AlphaZero, we rediscovered the hard way which components of the system were absolutely necessary to get right, and which components we could be messy with. It stands to reason that without the benefit of pre-existing work, monitoring systems are even more important in the discovery process. We discuss in particular,

Preprint. Work in progress.

the difficulties involved in generating, shuffling, and training on large datasets, as well as examples demonstrating why certain metrics were useful to monitor.

Minigo uses only publicly available services, APIs, and cloud offerings, and its source code, training artifacts, selfplay games, and evaluation games are publicly available at github.com/tensorflow/minigo. Additionally, various metrics and graphs are available at cloudygo.com

## 2   Related Work

A number of other efforts have been made to reproduce AlphaGo Zero. ELF OpenGo [4] is an implementation by Facebook; Leela Zero [5] is a community-run project using crowdsourced computational resources, and KataGo [6] is a project attempting to reproduce AlphaGoZero-style training using minimal computational resources. SAI[7] explores a multi-komi variant of AlphaGo Zero on 7x7 Go. As far as the authors are aware, Minigo is the only full-fledged AlphaGo Zero reproduction attempt that can be run by anybody on public cloud infrastructure.

## 3   Description of the Minigo System

At the heart of Minigo is the reinforcement learning loop as described in the AlphaZero[3] paper. (See the Appendix for a full comparison of AlphaGo Zero, AlphaZero, and Minigo). Briefly, selfplay with the current generation of network weights is used to generate games, and those games are used as training data to produce the next generation of network weights.

In each selfplay game. Minigo uses a variant of the UCT algorithm as described in the AlphaGo Zero paper to select a new variation from the game tree. The neural network considers the variation and predicts the next move, as well as the likelihood that one player will win the game. These predictions are integrated into the search tree, and the updated statistics are used to select the next variation to explore. A move is picked by either taking a weighted sample (first 30 moves) or picking the most visited variation. This is repeated until the game ends, one player resigns, or a move cap is reached. The final selected move thus takes into consideration the other player's responses and possible game continuations, and the visit counts can be used directly as a training target for the policy network. Additionally, the final game result can also be used as a training target for the value network. Each game's data (position, visit counts, game result) is then used to update the neural network's weights by stochastic gradient descent (SGD), simultaneously minimizing the policy and value error.

The number of readouts (800 for Minigo and AlphaZero) invested into each move roughly determines the ratio of compute required for selfplay and training. Minigo uses  800 Cloud TPUs for selfplay and 1 Cloud TPU for training.

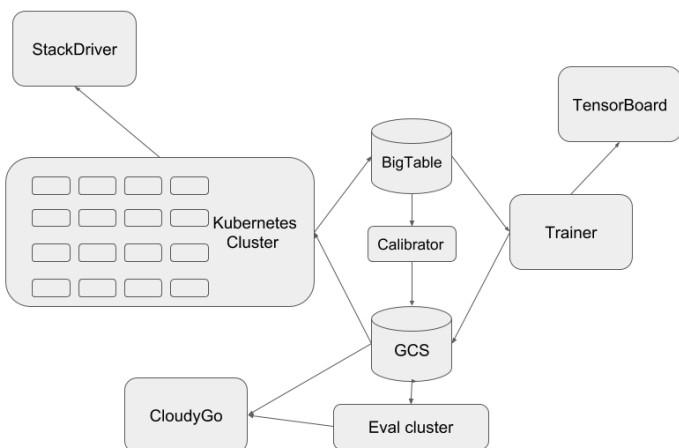

To orchestrate the many Cloud TPUs required for selfplay, we used Google Kubernetes Engine (GKE) to deploy many copies of a selfplay binary written in C++. Each selfplay worker writes training data directly to Cloud BigTable (CBT), with one row being one (position, visit counts, game result) tuple.

The trainer trains on samples from a sliding window of training data, and then publishes network weights to Google Cloud Storage (GCS). The selfplay workers periodically look for and download an updated set of network weights, closing the loop.

As part of our monitoring, a calibration job periodically processes game statistics and updates the hyperparameters used in selfplay, to ensure that the quality of selfplay data remains high. Additionally, we use StackDriver to monitor the health of the selfplay cluster and compute bulk statistics over all of the games being played. For example, we would log and track statistics about the distribution of value head outputs, winrates by color, distribution of game lengths, statistics about MCTS search depth & breadth, distributions of our time spent per inference, and the resignation threshold rate. We used TensorBoard to monitor the training job, keeping track of statistics like policy loss, value loss, regularization loss, top-1/top-3 policy accuracy, the magnitude of the value output, magnitude of weight updates, and entropy of policy output, all measured over the training selfplay data, held-out selfplay data, and human professional data. An evaluation cluster continually plays different generations against each other to determine their relative strengths.

Finally, we created a frontend (`https://cloudygo.com`) that would allow us to check various metrics at a glance, including data such as the relative ratings of each model, common opening patterns, responses to various challenging positions, game lengths, and the percentage of games that branched due to early game softpick. This frontend also served as a way to quickly spot-check selfplay games and evaluation games.

## 4 Building Minigo

We describe the evolution of the core components of Minigo, from project inception to their current state.

### 4.1 Self-play

#### 4.1.1 Initial prototypes

Minigo's selfplay is by far the most expensive part of the whole pipeline (100-1000 times as much compute as training), so many of our efforts were focused on improving the throughput of our selfplay cluster. We initially prototyped Minigo with a python-based engine, playing Go on a 9x9 board.[1] The size of this network (9 residual blocks of 32 filters) was easily evaluated without a GPU, and a small cluster of only a few hundred cores was able to keep up with a single P100 GPU running training. We used Google Kubernetes Engine to constantly run the containers, using its 'batch job' API to take care of scheduling, retrying, cleaning up dead pods, etc. Each docker container would start up, read the latest model from GCS, use it to play one game, and then write out the training data to a different directory in GCS, randomly picking either a training or holdout directory. This iteration of Minigo was able to start with random play and eventually reached a point where it was playing at a strong amateur level. However, given the lack of a professional human 9x9 dataset, and our lack of expertise in evaluating proper 9x9 play, it was difficult for us to understand how much (or how little) we had accomplished.

With our proof of concept in hand, we moved to a 19x19 version of the run, scaling up the depth and width of our network. It was immediately clear that we could not rely on a CPU-only implementation: although we initially moved to an intermediate-sized network for our 19x19 run, the full network would be 512 times[2] slower to evaluate than the network we had used for our 9x9 trial run. We started with 2000 NVIDIA K80 preemptible GPUs, making use of Google Kubernetes Engine to dynamically scale load. GKE additionally had an easy solution for managing the installation of NVIDIA drivers into many containers, so this saved us a great deal of trouble. We also implemented a vectorized MCTS that would minimize the Python overhead involved in tree search.

---

[1]The computational complexity scales to some large polynomial complexity at least $O(N^6)$: $N^2$ for the board area covered by convolution operations; $N^2$ for the approximate length of a game; $N$ for the depth of network required; $N - N^2$ for the number of reads in the game tree.

[2]Moving from a depth of 9 to 20 residual layers is 2x; moving from a width of 32 to 256 is 64x; and convolving over a 19x19 board instead of a 9x9 board is 4x, for a grand total of 512x

### 4.1.2 Batching techniques

It is useful to briefly describe several batching techniques that are available to us. As batching was the easiest way to scale up the throughput of our selfplay workers, it was the main driver behind various design decisions.

The easiest batching technique is to run multiple games in parallel. However, this comes with the drawback of increasing the latency of game completion: completing 16 games in 10 minutes is not necessarily better than completing 1 game in 1 minute.

An alternative batching technique is virtual losses in MCTS[8]. Virtual losses is a technique used to select multiple variations from a single MCTS game tree. To reiterate, MCTS is a tree algorithm that determines the best leaf to explore using a combination of probability priors (in this case from the 'policy' network) and the outcome of simulations from that node (in this case, from the 'value' network). Once the leaf is chosen, expanding it with an inference is slow, leaving an opportunity to do additional work. Unfortunately, MCTS deterministically identifies the 'best' leaf to expand, and has no notion of 'second best' leaf. To choose additional leaves, we can mark the originally chosen leaf as having been a loss, and then rerun selection to get another leaf. When the results of inference come back, the loss is replaced with the true evaluation. The number of simultaneous pending virtual losses is a tuneable parameter - increasing this number improves throughput but degrades selfplay quality.

### 4.1.3 Scaling up to Cloud TPUs

Using our K80 cluster, we ran the full Minigo pipeline 3 times as we worked out various bugs and added new infrastructure. However, we wished to see if the newly released Cloud TPUs (now in alpha on GKE) would yield improved performance. At the time, our Python engine had about 5% overhead on K80s, but we estimated that with TPUs, the python engine overhead would expand to 25% overhead or more. At this point, we rewrote our selfplay engine in C++, with an eye towards integrating MCTS very closely with Cloud TPU inference. With a modest number of TPUs (800), we could achieve a full run in about a week, assuming we could drive them near their theoretical limit.

For maximal throughput, Cloud TPUs demanded even larger batch sizes than what we'd been able to produce with virtual losses. To generate this throughput, we played multiple games simultaneously; each selfplay worker utilized one Cloud TPU to play 32 games in parallel at virtual losses = 8. Each game took 15 minutes on average, leading to about 2 completed games per minute per Cloud TPU.

With simultaneous games being played, we had to deal with the ragged edge problem - since not all games are the same length, when a game ended, we could either immediately start a new game, or suffer reduced throughput on the TPU. And if we continually started new games, what would we do when a new model was published? Our solution was to just switch models whenever a new one was published even if it happened in the middle of the game. We attempted to keep the pipeline balanced such that this happened only once a game, on average. Empirically, this appears not to have hurt the pipeline's performance.

One operational risk of this approach is that each resign threshold is calibrated to a particular model. If a new model significantly skewed value output compared to its predecessor, then many games near the resign threshold can all simultaneously resign. This would trigger a thundering herd problem as many games complete simultaneously and would possibly overload Cloud BigTable with heavy write volume. Worse still, since the resign threshold must be calibrated from resignation-disabled games (which are played to completion and therefore take 2-3x time to complete), entire games could be played with the wrong resignation threshold. To solve these issues, we set aside a small number of TPUs specifically for playing calibration matches with a much lower parallel game configuration, to minimize game completion latency. A ringbuffer was used to store the most recent calibration results and compute the threshold, and the buffer size kept small to ensure rapid updates.

With these changes, and with the help of Kubernetes, we were able to spin up over 100 petaops of compute, and flexibly and efficiently shift them around the load patterns in Google's cloud. The cluster of workers was able to achieve about 1.3-1.5ms per inference, playing about 1.8M games per day with the 'full size' 20 block network.

## 4.2 Trainer

The training part of the Minigo system has much in common with many supervised learning tasks. The only difference is that instead of training on a predefined dataset, we train on a continually evolving dataset. As TensorFlow is tailored towards a fixed input pipeline configuration, we found it easiest to restart our TensorFlow training process every time we wanted to shift our sliding window of training data. Other reinforcement learning frameworks like TF Agents[9] address this problem by maintaining an in-memory circular buffer of training data, but due to our shuffling requirements (discussed below), this was not feasible for us. From a systems perspective, we also designed our trainer system to pause if our selfplay cluster slowed down or stopped for any reason - otherwise, the trainer would overfit on the same data, necessitating a rollback intervention.

Other than the frequent restarts, our trainer followed best practices for supervised learning. For example, we'd originally used a custom data storage format designed for optimized compression of Go game data. While this worked well for our single-machine prototypes, we rewrote our training code into separate input pipeline and model code with TensorFlow Datasets and Estimator to best take advantage of Google Cloud TPUs. By using TensorFlow Datasets instead of Python iteration, we could make use of TensorFlow's parallel, buffered I/O features to directly feed data to the Cloud TPUs, first by reading directly from GCS, then by reading directly from Cloud BigTable.

## 4.3 Shuffler

Shuffling turned out to be an unexpectedly tricky and important part of our pipeline. A major complicating factor for shuffling was that AlphaGo Zero specified a large trailing window of 5e5 games (roughly 1e8 positions or 2e12 bytes, assuming a featurization of 19x19x17 array of float32) from which positions should be uniformly sampled. Ideally, sampling should be uniform; every SGD minibatch should be uniformly drawn from the entire dataset. Such a large dataset can be sharded to different degrees; if there are many smaller files, then uniform sampling requires the overhead of opening many different small files for each minibatch, and if there are a few large files, there is an overhead associated with seeking to the right position. (While each of our positions were fixed-length, TensorFlow's TFExample format is variable-length, so TensorFlow's APIs did not allow jumping to a specified offset.)

In practice, an approximation to uniform shuffling is required. At different points in Minigo's history, we used different approximations, each of which had different drawbacks. We found that every time we improved our shuffling, Minigo's strength would improve dramatically. The AlphaZero paper also reports that without using the 8-fold symmetry of the Go board to augment the training data, about 10x as many games needed to be played to reach the same strength. We believe that Minigo's sensitivity to proper shuffling arises from Go's gameplay pattern of placing stones sequentially. At a high level of play, stones are placed and rarely removed from the board, and thus specific subpatterns will persist through an entire game. Since every position will share the same training target of +/- 1 for the value network, it is easy for the neural network to overfit by simply memorizing each position.

We first noticed our lack of adequate shuffling when Minigo became extremely overconfident about games it played, as measured by the magnitude of its value network output in games against humans. We had also noticed that Minigo's performance on predicting outcomes of human professional games had dropped, but it was difficult to understand whether this was due to human professionals not playing perfectly or because Minigo was overfitting its value head. We obtained a conclusive answer when we started setting aside 5% of our selfplay games for validating our policy and value accuracy. Our network showed excellent performance on selfplay games which it had trained on, but near-random performance for selfplay games it had not trained on.

All in all, we learned that to shuffle effectively, we needed to adequately scramble each source of correlation in our data. We had two primary sources of correlation: intra-game correlation (every position from the same game shared the same result), and generational correlation (every game played by a given set of network weights would be of a similar 'style').

In our early shuffler implementations, we started from hundreds of thousands of tiny files, containing 100 to 1000 positions each. These files represented the raw output of thousands of selfplay workers as soon as they completed playing a set of games. Our first implementation, consisted of reading sequentially through the last million games, sampling 2% of positions, and emitting a series of chunks of 2048 positions each. We would then train on these chunks in shuffled order. This method failed

on both shuffling criteria: each chunk contained on average, more than 1 position from the same game, and each chunk consisted of games from the same generation. To fix both issues, we utilized a machine with 64GB memory to perform a perfect shuffle on positions uniformly sampled from the last 5e5 games. Even then, we observed a large boost in strength when we randomly additionally applied one of 8 symmetries of the Go board to each selected position.

This shuffler implementation served us up until we replaced our cluster of K80 GPUs with Cloud TPUs. When this happened, our single-threaded shuffler reading TFExample files from GCS became the new bottleneck. We patched our implementation with a Python multiprocessing pool, which bought us enough time to implement a Cloud BigTable solution. With CBT, our selfplay cluster would directly write one row into CBT for each position.

In our Cloud BigTable pipeline, each TFExample was written to its own sequential row, with an individual row index determined by its game number and move number within the game, so that each individual move in the entire list of games could be randomly accessed. The sequential ordering was necessary because CBT's scan range operations are lexicographic, and we needed to be able to sample moves from a predictable game range (the last N games played by the cluster).

Cloud Bigtable provides probabilistic sampling as part of its server-side API. The lower the probability, the better the savings in bandwidth, since only the selected rows are transferred. Computing the correct sampling probability required some bookkeeping, since the number of moves in a game can vary, and there is no method to count the rows in a CBT lexicographic row range aside from iterating through them. Therefore, the move count per game was stored in a separate keyspace in the table, so that for any given game range, the total moves in that range could be calculated, and from that total, the correct sampling probability (typically around 1.5%).

We then executed a final in-memory shuffle of the sampled row keys before requesting the row keys from CBT. Training ran in parallel with consumption of shuffled data, decreasing the overall training time by 50%, and removing the input pipeline as the bottleneck.

## 4.4  Calibration

Another unexpectedly tricky part of the Minigo pipeline was our calibration process. The AlphaGo Zero paper described using a resignation threshold to terminate selfplay when both the current position's evaluation was heavily in favor of one side. This early termination is valuable in saving compute, as Go has an onerous game-end condition requiring playing the game out for many hundreds of moves. [3] To determine this resignation threshold, the AlphaZero paper specified that 10% of games should be played to completion, and a threshold chosen such that fewer than 5% of games would have been incorrectly resigned.

We discovered that this early termination had several second-order effects on the Minigo system. The most severe consequence was that an incorrectly calibrated resignation threshold could destabilize training by creating a pessimistic loop when a game was prematurely resigned. When that resignation was subsequently used as training data, it would be even more likely that the network would resign if presented with the same position. This only occurred occasionally, usually due to a different bug, but such a pessimistic loop would cause training to diverge. In order to detect this condition, we tracked statistics on winrate by color, distribution of game lengths, and the average magnitude of the value network's output during selfplay. A pessimistic loop would typically result in a heavy skew of winrate, drastically shorter games, and a very confident value network output but a very high value error on holdout games.

To avoid a pessimistic loop scenario, we started to calibrate our resignation threshold to a more conservative 3% false positive rate, even if it meant playing longer games. Additionally, we invested in lowering the latency involved in computing the resignation threshold. Originally, our resignation threshold was computed by running a custom script over the most recent resignation-disabled games. This script was executed by hand, and the resignation threshold configuration propagated by pushing a new flagfile to GCS. Being human dependent meant that the script was not run quite as often as it should have been, leading to our threshold being fairly conservative and having the false positive rate under 5%. Eventually, with our Cloud BigTable rewrite, we could compute the resignation threshold

---

[3] In games played by humans, a game is "over" when both players agree on the status of all territory and stones. Disagreements are resolved by continuing the game, explicitly capturing all stones. Computers usually need to play a game to its completion to avoid any ambiguities.

by directly querying CBT. We updated our configuration mechanism to be more lightweight and automated the calculation, and the lowered latency improved the robustness of the early stages of our pipeline, where the network was rapidly learning the basics of the game.

Another consequence of early termination was that the distribution of training data shifted towards early- and mid-game examples. Amusingly, this meant that our network would make basic mistakes in end-game positions when we ran test matches against humans, because there were almost no end game positions in the training data. We discovered this by posting our bot on an online Go server, where its human opponents had no preconceived notions of what the bot was "supposed" to do. We also set up a BigQuery dataset of moves played, and crafted a SQL query that would detect instances of games where the estimated winrate was 99%+ in favor of one side before suddenly flipping to 99% in favor of the other side. To fix our end-game deficiency, we also trained on the resignation-disabled games so that there would be end-game positions to learn from.

## 4.5   Evaluation Cluster

We allocated a small cluster of GPUs to play evaluation games between different models. Evaluation games disabled all of the settings that encourage exploration, like dirichlet noise and temperature. By using a modified Bradley-Terry system as implemented by the Python choix package, we computed a single parameter rating which we convert to the more familiar Elo number for display purposes.

Measuring the relative strength of our models was useful for several reasons: - To figure out when a run has stopped improving, - To find the strongest models in a run. (Without gating, strength is not guaranteed to improve between each model, and in practice it was observed that successive models could have rating swings of up to 900 Elo.) - To compare our rating-over-time graphs to those shown in the AG / AGZ / AZ papers.

Our evaluation methodology differed in one important respect from the methodology described by AlphaGo Zero. There, evaluation was used as a method of gating the promotion of new models, and each model would only play against the currently active model. This methodology potentially suffers from transitive cycles where A beats B; B beats C, and C beats A. Our evaluation methodology used a more diverge array of opponents, playing model N versus models N-1, N-2, N-3, N-5, N-10, N-20, and N-50. Thus, as the run proceeded, each model would play a symmetric set of pairs of models that came ahead and before it. Additional games were chosen to reduce the uncertainty in each model's rating for models that had not played similar strength models.

As we experimented and made improvements, it became important to assess models from different runs. Cross-run evaluation games were played between the strongest models of each run to determine the impact of our hyperparameter changes and pipeline bug fixes on overall strength. We obtained the most reliable cross-run ratings by taking the best models at regular intervals from each run, and then playing an equal number of games for all n-choose-2 pairings.

Beyond the use in evaluating training runs, the evaluation cluster was used for a number of one-off tests, such as tuning various hyper parameters (pUCT, virtual loss size, Q initialization), testing against other networks trained via completely different approaches, (e.g. supervised, transfer learning, or alternate AlphaGo reproductions), and playing matches against external reference points.

## 5   Lessons for Reproducibility of Reinforcement Learning

In reproducing AlphaGo Zero, we found it useful to monitor nearly everything we could think of, to prove to ourselves that our implementation was bug-free. We found Figure 3 from the original AlphaGo Zero paper to be a very useful guidepost, and wish that we had more metrics to compare against - a richer array of training metrics on e.g. policy entropy, value confidence, policy confidence would not be overly burdensome to provide, and would have greatly aided our reproducion efforts. And even if the raw data were not provided, it would have been valuable to know which metrics we should monitor.

We also found that healthy metrics were necessary but not sufficient for success. Our evaluation cluster was the only true indicator for success, and was of tremendous help in letting us know that a change in our hyperparameter settings had had a positive effect.

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

# Appendix

Table comparing hyperparameter choices between AlphaGo Zero, AlphaZero, and Minigo

|  | AlphaGo Zero | AlphaZero | Minigo |
|---|---|---|---|
| Reads during Selfplay | 1600 | 800 | 800 |
| cPUCT | ? | ? | 1.7 |
| Training algorithm | SGD, momentum=0.9 | same | same |
| Learning rate | 1e-2, 1e-3, 1e-4 | 2e-2, 2e-3, 2e-4 | 1e-2, 1e-3, 1e-4 |
| Learning rate cuts at step | 4e5, 6e5 | 3e5, 5e5 | 4e5, 6e5 |
| L2 regularization weight | 1e-4 | 1e-4 | 1e-4 |
| Promotion criteria | 55% win against previous | Always | Always |
| Training window size | 500,000 games | 500,000 games | 500,000 games |
| Games per checkpoint | Unknown | Unknown | 12,000-25,000 |
| Virtual loss parallelism | Unknown | Unknown | 8 |
| Initialization of Q | Init to loss | Init to loss | Init to loss |

