# OpenReview forum: "Minigo: A Case Study in Reproducing Reinforcement Learning Research"
_ICLR.cc/2019/Workshop/RML — RML 2019_

### Official Review · AnonReviewer1 · 2019-04-01
**Interesting Work**

**Rating:** 5
**Confidence:** 3

**Review:**

I enjoyed reading the paper. I wish there are more papers which study how to debug a model which requires lots of computational resources.

This paper focuses on difficulties involved in models which uses  large datasets for training. This paper makes the claim that in order to understand better what the model is doing, it's important to have monitoring systems in the discovery process.
This paper does a pretty good job in describing the components involved in a system like
AlphaGo (and its variants). Specifically, paper focuses on reproducibility in self-play algorithms.
Its interesting to know that conclusion of the paper is  to monitor nearly everything one could think of, to make sure that the implementation is bug-free.

---

### Decision · Program_Chairs · 2019-04-05
**Acceptance Decision**

Accept